# No-Restraint Committed General Hospital Psychiatric Units (SPDCs) in Italy—A Descriptive Organizational Study

**DOI:** 10.3390/healthcare12111104

**Published:** 2024-05-28

**Authors:** Raffaella Pocobello, Francesca Camilli, Giovanni Rossi, Maurizio Davì, Caterina Corbascio, Domenico Tancredi, Alessandra Oretti, Tommaso Bonavigo, Gian Maria Galeazzi, Oliver Wegenberger, Tarek el Sehity

**Affiliations:** 1Institute of Cognitive Sciences and Technologies, National Research Council, 00196 Rome, Italy; 2Club SPDC—No Restraint, 34143 Trieste, Italy; 3Dipartimento Transmurale Salute Mentale, Azienda Provinciale per i Servizi Sanitari—APSS, 38100 Trento, Italy; 4Dipartimento Dipendenze e Salute Mentale, Azienda Sanitaria Universitaria Giuliano Isontina, 34128 Trieste, Italy; 5Department of Biomedical, Metabolic and Neural Sciences, University of Modena and Reggio Emilia, 41124 Modena, Italy; 6Department of Mental Health and Drug Abuse, Azienda USL-IRCCS di Reggio Emilia, 42122 Reggio Emilia, Italy; 7Faculty of Psychology, Sigmund Freud Private University, 1020 Vienna, Austria

**Keywords:** human rights, No restraint, mechanical restraint, coercion in mental health care, involuntary hospitalization, general hospital psychiatric units

## Abstract

This study describes and explores the application of no-restraint policies in General Hospital Psychiatric Units (GHPUs) in Italy, a country pioneering in deinstitutionalization and psychiatric reform. The research aims to assess the organizational characteristics and effectiveness of no-restraint practices, contributing to the global discourse on humane psychiatric care. Following a purposive sampling approach, a nationwide descriptive study was conducted involving a detailed online survey distributed to 24 GHPUs actively engaged in or aspiring toward no-restraint practices. The survey, comprising 60 items across seven sections, gathered comprehensive data on the structural, organizational, and operational dimensions of the units, along with the prevalence and management of restraint episodes. Results reveal a significant commitment to no-restraint policies, with 14 GHPUs reporting zero restraint incidents in 2022. Despite variations in infrastructure and staffing, a common thread was the implementation of systematic procedures and risk management training aimed at reducing coercive practices. The study identified a correlation between the use of exclusive garden spaces and an increased incidence of restraints, suggesting nuanced factors influencing restraint practices. The findings underscore the viability and ethical alignment of no-restraint practices within psychiatric care, highlighting the crucial role of organizational protocols and training. This research adds empirical weight to the advocacy for restraint-free environments in mental health settings, signaling a paradigm shift toward more humane and rights-respecting psychiatric care.

## 1. Introduction

Recent global initiatives, notably the Convention on the Rights of Persons with Disabilities (CRPD) by the United Nations and the Council of Europe’s Resolution 2291, have advocated for the cessation of coercive practices in mental health care [1,2]. The World Health Organization further supports this movement by advocating for a human rights-based approach and calls for increased research into effective strategies to avoid the use of coercion [3]. Despite some successes, significant research and application gaps remain [4]. The EU-COST Action ‘FOSTREN’ addresses these gaps, examining ways mental health services can eliminate coercive methods [5].

In Italy, the ‘No restraint’ initiative has been central to the deinstitutionalization process since its inception nearly fifty years ago, emphasizing the ethical treatment of patients and the elimination of physical restraints in psychiatric care. Franco Basaglia, a key figure in Italian psychiatric reform, was pivotal in this movement. As director of the psychiatric hospital in Gorizia, Basaglia famously declared, “Mi no firmo” (Venetian dialect for “I do not sign”) when asked to authorize the use of restraints on patients. At the time, such authorizations were routine and required the medical director’s signature. Basaglia’s refusal highlighted his ethical opposition to what he considered an inhumane practice.

Basaglia’s stance against coercive methods became a cornerstone of the broader deinstitutionalization movement, which advocated for a more humane and supportive approach to mental health care. By rejecting restraint, Basaglia and his colleagues paved the way for community-based services that prioritize patients’ human rights.

This study examines a notable practice developed within the community mental health service network in Italy, known as the ‘Open door—No restraint’ policy [6], which aligns with FOSTREN initiatives. This approach especially affects the general hospital psychiatric units (GHPUs), which in Italy are called SPDCs—i.e., *Servizio Psichiatrico di Diagnosi e Cura* (in English Psychiatric Diagnostic- and Treatment Services). In Italy, GHPUs are located within hospital facilities but are an integral part of a Mental Health Department (MHD), which also includes other facilities, such as community mental health centers and day care centers. Following a model of psychiatric care that is based on community services, national directives indicate that each GHPU should contain no more than 16 beds and should be provided with common spaces [7]. Although this indication is not binding and the autonomy of the regions determines some variability in the actual number of beds per GHPU, there were 323 active GHPUs providing 3897 beds for standard admissions and an additional 282 for day hospital services (average number of psychiatric beds per GHPU: 12.9) in Italy in 2022 [8].

GHPUs that are inspired by the ‘No restraint’ policy follow (or strive to follow) two main principles: (1) The practice of never using physical restraints, including bed restraints; (2) The practice of keeping the doors of the ward open. This approach emphasizes the relational aspects and promotes the responsibility of people at the center of care while respecting their freedom and dignity. The implementation of the ‘No restraint’ practice has developed starting from the ‘Trieste Model’ but has been enriched over the years by various experiences, which have unfolded in different ways and contexts. At the moment, the ‘Open door—No restraint’ is not an ‘institutionalized’ approach but rather a concrete practice inspired by certain shared principles [6].

The ‘Club Spdc No Restraint’ Association is a network that gathers GHPUs following or inspired by the ‘No restraint’ policy [6]. Representatives from these GHPUs have started to connect and exchange their experiences since 2002, leading to the official foundation of the association in 2013. The ‘Club Spdc No Restraint’ regularly collects information on the structural and organizational characteristics of the represented GHPUs—currently at 24—and keeps track of the timing and numbers of mechanical restraint episodes. The association accepts as members not only representatives from GHPUs that have reached zero restraint but also those working toward it in order to support them in their mission. Accordingly, when we use the expression ‘No restraint committed GHPUs’ in this article, we refer both to GHPUs that already follow the ‘No restraint’ policy and to GHPUs that are inspired by these principles and are seeking to improve their practice.

Although ‘No restraint’ initiatives may considerably impact the quality of psychiatric care, comprehensive research on the topic has yet to be published. Results from the PROGRES-Acute project (*PROGetto RESidenze*, Residential Care Project for Acute Patients) indicated that 18 out of 262 GHPUs never used ‘locked doors’ in 2002–2003 [9]. However, since these data were collected more than two decades ago, they may not represent the current situation. In one of the few published works documenting ‘No restraint’ practices, Davì provided an overview of the first four years of the ‘Open doors experience’ in Trento, Italy, describing the operational procedures followed in the GHPU and reporting some quantitative outcomes, such as absconding, aggressive acts, client satisfaction index, staff emotional burden, and related formative needs [10,11]. More recently, Zanfini and colleagues extensively described the experience of the ‘No restraint’ GHPU in Ravenna by identifying four main pillars, i.e., the organization of the spaces, the use of registries and audits, clinical and care activities, and staff training [12]. On the other hand, Montanari Vergallo and Gulino highlighted a ‘discrepancy’ between legal and ethical recommendations and organizational conditions in Italy, mainly because of a lack of funding and specific training programs [13].

On this issue, it has not been mentioned that, in 2022, the Italian Ministry of Health allocated 60 million euros to projects aimed at improving the services of the MHDs. One of the explicitly mentioned goals was the implementation of regional projects aimed at “overcoming mechanical restraint” [14]. Expected actions include initiating/implementing pathways such as training courses for professionals, initiatives aimed at monitoring restraint episodes, and protocols/guidelines to guarantee the quality of service facilities. Data for the regional projects have not yet been made available.

On the other hand, the possible impact of organizational factors, such as staff and ward characteristics, on coercion has also been investigated at the international level [5,15,16,17,18,19,20]. The number of staff members per service user, the male–female staff ratio, the frequency of use of substitute staff, the variability in teams’ work experience, and staff education have all emerged as possible factors that could have an impact on reducing or eliminating coercive practices [16,17,18]. The presence of outdoor space, special safety measures, the number of service users in the building, total private space per user, and level of comfort have been investigated as ward design features influencing the use of seclusion [20]. The ward’s size and location (i.e., rural areas or small towns vs. metropolitan areas) have also been related to different seclusion or restraint rates [15,19].

The lack of comprehensive scientific data and the experience reported by the ‘Club SPDC No Restraint’ Association suggested that an investigation at the national level could provide a significant overview of the status of the ‘No restraint’ practices in Italian GHPUs. Therefore, the present study aimed to collect and analyze data on the current experiences of the Italian ‘No restraint’ committed services.

More specifically, the main research objectives were the following:Document the experience of ‘No restraint’ committed GHPUs.Identify whether different models of ‘No restraint’ committed GHPUs currently exist in Italy.Provide an overview of ‘No restraint’ committed GHPUs and describe the organizational characteristics of the identified models.

## 2. Methods

### 2.1. Design

We carried out a census study to investigate how members of the ‘Club SPDC No Restraint’ Association in Italy implement their ‘No restraint’ policies. An online survey was conducted for the collection of data on the structural and organizational features of the GHPUs, the number of restraint episodes (if any), the numbers and roles of the professionals active in the GHPUs, their activities, and the monitoring carried out.

### 2.2. Instrument

We adopted a comprehensive survey instrument developed by the ’lub SPDC No Restraint’.

The questionnaire was initially developed by Giovanni Rossi and subsequently revised by the ‘Club SPDC No Restraint’. This initial development ensured that the instrument was grounded in practical, real-world insights from professionals directly involved in the field.

The final iteration of the questionnaire underwent further refinements from the research team at the ISTC-CNR, which included experts in mental health services, survey methodology, and statistical analysis. Specific modifications included refining the language for clarity, adding questions to capture additional relevant variables, and restructuring sections to improve the flow and coherence of the questionnaire. Each of these changes was made based on feedback from preliminary pilot testing and expert reviews to ensure the instrument’s validity and reliability.

The questionnaire comprised 60 items divided into seven sections, designed to gather detailed, descriptive, and objective data across various aspects of GHPU operations. These sections included informant identification, facility-specific data, data on restraint utilization, infrastructure, staffing details, activities, monitoring protocols, catchment area, and departmental organization, along with an open-ended section for additional comments. This structured approach was intended to minimize personal biases and enhance the reliability of the data collected. The questionnaire is linked to this article in the Appendix A section.

### 2.3. Ethical Considerations

The study received the ‘Ethical Clearance’ approval (Prot. no. 0199902) from the Ethical Committee of the National Research Council of Italy. When opening the link to the online survey, participants were prompted to provide informed consent, which clarified the aims of the study, the procedures, as well as the risks and benefits of participating in the research. Respondents were also shown a second text with information regarding data processing. Participants were asked to fill in the survey only if they gave both consents (i.e., “Yes, I agree to participate in the above-mentioned study” and “Yes, I consent to the processing of my personal data”).

The survey did not gather any personal data from the respondents except for those required for the purpose of collecting their informed consent. Personal data were stored separately from the data of the GHPUs, and their access was limited to the ISTC-CNR research team. Participants provided their working contacts (email and phone number of the psychiatric ward), which have been used to clarify some data.

### 2.4. Participants and Data Collection

Following a purposive sampling approach, the initiative was disseminated among the 24 GHPUs that were known to embrace the ‘No restraint’ approach due to their membership at ‘Club SPDC No Restraint’. More specifically, representatives of the GHPUs were contacted via email by G.R. (President of the Association) and received the link to fill in the online survey. In some cases, two or more professionals of the same GHPU received the invitation email.

The survey was carried out through the Lime Survey platform (https://www.limesurvey.org/) from April to June 2023. After that date, the survey was automatically deactivated, and we exported the data into Excel format. All the 24 GHPUs who received the invitation participated in the survey. For one of the GHPUs, data were inserted two times, with a few minor inconsistencies between the two records. The respondent was contacted via email, and a phone interview was conducted to clarify these details.

### 2.5. Data Diagnostics

Data were examined and adjusted for consistency. Where feasible and available, missing data were supplemented by recontacting survey respondents through email. Out of the 24 questionnaires collected, 12 services had to be recontacted due to incomplete datasets so that a complete dataset with no missing values was created.

### 2.6. Data Analysis Strategy

The survey’s descriptive data were verified for consistency using Excel 2019 spreadsheets and subsequently imported into SPSS^®^ 27.0 and Jamovi (Version 2.3.28) for descriptive and exploratory analyses [21,22]. In the descriptive analysis, continuous variables were summarized using means (M) and standard deviations (SD), while proportions were used for discrete count variables. Pearson chi-square test was used to determine the significant association between two categorical variables; a significance threshold was set at *p* < 0.05 for all analyses.

Negative binomial regression analysis was applied to explore and identify how predictors and related factors influence the counts of restraints since such counts tend to vary more widely than their average values (over-dispersion).

A Multiple Correspondence Analysis (MCA) was conducted to explore the relationships among various categorical variables reported in the survey. As described by Hjellbrekke [23], MCA is an explorative approach used to visualize the relationships in multivariate categorical data as represented in a contingency table. This analysis is particularly relevant for examining patterns and associations between multiple categorical variables. This technique uses chi-square distances to project the data into a space where the relationships between variables can be easily understood in terms of their similarity or dissimilarity. MCA also differentiates between active and supplementary variables, with the latter not influencing the dimension definitions but still being positioned close to the categories that best describe them.

## 3. Results

A total of 24 GHPUs in Italy participated in the study whose informants (mostly directors) confirmed the commitment of their GHPU not to use restraints. Figure 1 provides a synopsis of the geographic locations of the GHPUs described in this study.

### 3.1. Organisation of the MHDs and Catchment Areas

On average, the GHPUs had target populations of 296,710 inhabitants (SD = 167,099), ranging from 108,000 to 780,000. The target population of the whole MHD ranged from 150,000 to 1,602,000 inhabitants (M = 573,540; SD = 320,138). It was reported that one MHD included, on average, about 2.7 GHPUs (SD = 2.1), with a range from 1 to 9 GHPUs. Most GHPUs (83.3%) were under the control of a health authority (‘azienda sanitaria’), while one GHPU was part of a hospital organization (‘azienda ospedaliera’). Three GHPUs (12.5%) were indicated to be part of ‘other’ types of organizations (e.g., integrated university hospitals).

### 3.2. Facilities Descriptors

On average, each GHPU had 12.6 (*SD* = 4.6) beds, ranging from 4 to 24, with a surface area ranging from 180 m^2^ to 1134 m^2^ (*M* = 546 m^2^, *SD* = 282 m^2^). Two-thirds of the GHPUs (66.7%) had at least one single room, and almost all GHPUs (91.7%) had double rooms. More than half of the GHPUs had a room structure other than single or double rooms (e.g., triple rooms, quadruple rooms, or rooms that can be adapted according to the needs). Roughly half of the GHPUs (45.8%) were located on the ground floor. In nearly all GHPUs (95.8%), maintenance and cleaning were perceived as appropriate to keep the environments decent.

Many GHPUs (79.2%) had an alarm bell present in every single room, and 75% of the GHPUs had ensuite bathrooms in all rooms. Nearly all GHPUs had a dining room (95.8%), and all GHPUs had a living room with a TV; a smoking environment with functioning exhaust fans was present in two-thirds of the GHPUs (66.7%). Slightly more than one-third of the GHPUs (37.5%) had a drink dispenser, nearly one-third (29.2%) had no drink dispenser but another service (e.g., cafeteria), and one-third (33.3%) had no drink dispenser and no other service. An armored access door was present in less than half of the GHPUs (37.5%). Two-thirds of the GHPUs had no cameras present (66.7%); of the eight GHPUs that had cameras, five had cameras only in some rooms. Some kind of garden or outdoor space was present in most GHPUs (75%; see Table 1 for a detailed description).

### 3.3. Restraint Practices

Of the 24 GHPUs involved in the study, 14 (58%) reported zero restraint episodes between 1 January 2022 and 31 December 2022. In 10 GHPUs, two or more restraints took place (median of 4.5 restraints ranging from 2 up to 43 restraints; see Figure 2 for an overview).

Ten GHPUs reported a total of 101 restraints during the twelve-month period in 2022, with 73% of these restraints lasting less than 24 h. One GHPU reported an outlier with a duration of 14.5 days; the shortest restraint lasted 10 h (see Figure 2).

When evaluating which standards are maintained in case of mechanical restrictions, we observed that 9 out of 10 (90%) GHPUs ensure the continuous presence of a professional at the user’s side (standard A) and/or initiate a multiprofessional and multi-service audit (standard B).

### 3.4. Staffing and Shift Distribution

Medical doctors and nurses were employed in all GHPUs surveyed (*n* = 24/24, 100%). The median number of medical doctors per GHPU was four, with a range from one to 14, while nurses were represented with a median number of 14.5, with a broader range from nine to 29.

A smaller proportion of 45.8% (*n* = 11/24) GHPUs employed psychologists, none reporting more than six psychologists. Care workers were present in 91.7% of GHPUs (*n* = 22/24), with each GHPU having up to 12, though some reported none.

Psychiatric rehabilitation technicians and social workers were present in less than a third of the GHPUs (29.17% for each profession, *n* = 7/24), with a reported range of zero to six and zero to four, respectively. ‘Other professions’ were reported in 33.3% (*n* = 8/24) of the GHPUs, with GHPUs reporting up to two staff members being trained in other professions (see Table 2).

Nearly all GHPUs reported having a psychiatrist on duty or on call during the nights (*n* = 23/24, 95.83%). In most GHPUs, medical doctors from the MHD performed night shifts on duty in the GHPU (*n* = 21/24, 87.50%).

Likewise, the shift-wise perspective on the professionals employed across GHPUs data indicated a high presence of medical doctors and nurses across all shifts. The presence of psychologists and social workers was more limited and primarily confined to daytime shifts. Psychiatric rehabilitation technicians and other unspecified professions had varied but generally lower presence across shifts (see Table 3).

### 3.5. Activities and Monitoring

On average, service users in the GHPUs spent 379.81 days (SD = 457.93) on Commitment and Forced Treatment, with a range from 7 to 1682 days (note: three GHPUs did not respond to this question). The number of days spent on Commitment and Forced Treatment represents, on average, 15.4% (SD = 11.1%) of the total number of days of hospitalization. The range is 4.5% to 50.0% (note: four GHPUs did not respond to this question).

Almost all GHPUs monitored injuries due to aggression toward professionals (*n* = 23/24, 95.8%). Scales and questionnaires, including outcome assessment, were used in a little more than half of the GHPUs investigated (*n* = 14/24, 58.3%). A third of the GHPUs admitted service users above the maximum registered bed number (*n* = 9/24, 37.5%). Medical, nursing, and therapy records were computerized in roughly half of the GHPUs (*n* = 14/24, 58.3%). Most GHPUs used a procedure for self- and hetero-directed aggressive risk management (*n* = 19/24, 79.2%). Many GHPUs had access and discharge procedures with MHD interfaces (*n* = 22/24, 91.7%). The same applies to the existence of a procedure in the Emergency Room with respect to consultations and admissions, which applies to *n* = 22/24 GHPUs (91.7%).

Ward rounds, individual and group psychological interventions, and psychosocial interventions for family members were carried out in about half of the GHPUs. Nearly every GHPU carried out individual outpatient interviews (Table 4 shows detailed statistics). Additionally, psychoeducation, expressive and recreational activities, rehabilitation groups, network interventions, multiprofessional evaluation meetings, supervision, recovery-oriented interviews, recovery groups, resocializing and rehabilitative activities, animal-assisted interventions, mindfulness, recreational activities were mentioned by single GHPUs as activities being carried out.

Most GHPUs were using individual care plans (*n* = 22/24, 91.7%) and private spaces for the individual administration of therapy (*n* = 20/24, 83.3%). Half of the GHPUs had a procedure with the local police for interventions in case of Commitment and Forced Treatment or emergency in the ward (*n* = 12/24, 50.0%). In all GHPUs, service users could keep personal items such as telephones, computers, etc. (*n* = 24/24, 100%). Half of the GHPUs allowed inpatients to go out alone (*n* = 12/24, 50.0%). The majority of GHPUs have implemented a staff training program on self- and hetero-aggression risk management on a regular basis (*n* = 19/24, 79.2%).

GHPUs that carried out one or more restraints in 2022 (*n* = 10/24) documented each restraint in a register. In most cases, procedures to eliminate the use of mechanical and spatial restraints were reported to be present. Furthermore, restraint episodes were subject to audits or periodic reviews, and continuous staff assistance was provided in the case of restraint in most GHPUs. A procedure to reduce the risk and duration in case of restraints was present in half of the cases (Table 5 shows detailed statistics).

### 3.6. Exploraty Data Analysis toward the Description of No-Restraint GHPUs

#### 3.6.1. GLM Analysis

A Spearman’s rho correlation analysis was conducted to explore the relationships between the total number of people subject to restraint during the year and potentially meaningful predictor variables for a GLM regression analysis focusing on the number of restraint inpatients within facilities. The results indicated a significant positive correlation between the total number of inpatients subject to restraint during the year and the presence of a garden or outdoor space exclusive to the GHPU (*ρ* = 0.451, *p* = 0.027), suggesting that facilities with a garden or outdoor space exclusive to the GHPU tend to have higher numbers of people subject to restraint. Additionally, a significant negative correlation was found with the use of scales and questionnaires, including outcome assessment (*ρ* = −0.525, *p* = 0.008), indicating that the more these tools are used, the fewer the number of people subjected to restraint. There was also a significant negative correlation with the availability of a procedure for self- and hetero-directed aggressive risk management (*ρ* = −0.488, *p* = 0.015), suggesting that the presence of such a procedure is associated with fewer instances of restraint.

The mean of the count variable (number of inpatients restrained) is *M* = 2.21, while the variance is about 14.09, indicating overdispersion with a variance-to-mean ratio of 6.38. Consequently, a negative binomial generalized linear model was conducted to examine the effects of the presence of a garden (exclusive use for GHPU), the use of scales for outcome assessment, and the existence of a procedure for self- and hetero-directed aggressive risk management on the number of users restrained. An interaction term between the type of GHPU and the use of scales and questionnaires was also included in the model. The link function used was log, which is suitable for counting data with overdispersion, as indicated by the negative binomial distribution.

Model Fit and quality were good with an R-squared of 0.703, indicating that approximately 70.3% of the variance in the dependent variable is explained by the model; AIC: 76.067, BIC: 86.314, and Deviance: 23.657, suggesting the model’s relative quality in balancing goodness-of-fit with complexity; chi-squared/DF: 1.341, pointing to overdispersion in the data, which justifies the use of a negative binomial model.

Table 6 reports the effects on the number of restrained users in GHPU (see Table 6). Significant effects were found for the type of GHPU—open or closed doors policy (*χ*^2^(1) = 4.79, *p* = 0.029) and the garden for the exclusive use of the GHPU (*χ*^2^(1) = 16.59, *p* < 0.001), with parameter estimates indicating that having a garden with exclusive [excl. GHPU], is associated with an increase in the number of inpatients restrained (*B* = 1.77, *p* < 0.001). A marginal effect was observed for the procedure for self- and hetero-directed aggressive risk management (*χ*^2^(1) = 3.07, *p* = 0.080).

The parameter estimates indicated that the use of scales and questionnaires (*χ*^2^(1) = 7.81, *p* = 0.005) is associated with a decrease in the number of inpatients restrained (*B* = −1.98, *p* = 0.008). This is even more so when considering the interaction effects with the type of department (*χ*^2^(1) = 4.11, *p* = 0.043).

Departments’ open doors policies (*B* = −1.46, *p* = 0.038) and the use of outcome measures (*B* = −1.98, *p* = 0.008) are associated with a decrease in the number of users restrained with a significant interaction term (*B* = 2.07, *p* = 0.047) (see Figure 3), suggesting a differential effect of using scales and questionnaires on the number of users restrained, depending on the type of GHPU.

#### 3.6.2. Hierarchical Cluster Analysis and Multiple Correspondence Analysis

Hierarchical cluster analysis was conducted to examine the clustering of the 24 GHPUs based on their most informative dichotomized variables (I13; I25, I27, I37, I45, I51, I53, I54) using Ward’s method and the squared Euclidean distance measure appropriate for binary variables [24].

In the agglomeration schedule, a significant increase in the coefficient value indicates a substantial jump in dissimilarity when moving from one stage to the next, representing a good indicator of the optimal number of clusters. The jump from stage 20 to stage 21 (from 25.086 to 30.586; 4 clusters) suggested a considerable increase in dissimilarity. Another noticeable increase emerges from stage 22 to stage 23 (from 36.686 to 45.500; 2 clusters). The cluster membership output provides the assignment of cases for 2 and 4 clusters. The final analysis resulted in a dendrogram (see Figure 4) illustrating the clustering process based on which a four- and a two-cluster-solution emerged. When inspecting the agglomeration schedule, a substantial jump in dissimilarity can be seen when moving from one stage to the next. This jump can be a good indicator of the optimal number of clusters.

To facilitate the interpretation of the two and four clusters of GHPUs, we conducted a Multiple Correspondence Analysis (MCA), which summarizes the relationships among the categorical variables in our research. The GHPUs were mapped as supplementary variables; as a result, they were close to the categories that most accurately represented them.

The MCA resulted in two significant dimensions explaining 46.9% of the variance in the dataset, with the first and second dimensions accounting for 25.6% and 20.5% of the variance, respectively. The Cronbach’s Alpha values for dimensions 1 and 2 were 0.675 and 0.568, respectively, suggesting an acceptable level of reliability in the associations captured by each dimension (see Table 7).

Dimension 1: As visualized in Figure 5, the first dimension was strongly associated with variables related to restraint, risk management procedures, and protocols in GHPU. The highest correlation was observed with the item “No Restraints” (I13; *r* = 0.424), “Staff training programme on self- and hetero-aggression risk management implemented on a regular basis?” (I54; *r* = 0.447), and “Is there a procedure for self- and hetero-directed aggressive risk management?” (I40; *r* = 0.688) and “Do you use scales and questionnaires including outcome assessment?” (I37; *r* = 0.592).

Dimension 2: The second dimension appears to reflect the practices related to patient autonomy and the therapeutic environment. Items like “Garden is for the exclusive use of the GHPU” (I27; *r* = 0.417) and “Are in-patients allowed to go out alone?” (I53; *r* = 0.488) and individual administration of therapy in private spaces (I45; *r* = 0.543) show strong coordinates on Dimension 2, indicating their relevance to this dimension (see Figure 5).

The output of the MCA depicted in Figure 5 evidences that the four clusters of GHPU may be best captured according to the dominant categories in their proximity:-GHPUs of Cluster 1 may be prone to some restraint practices due to their garden infrastructure for the exclusive use of GHPU (I27) and the presence of armored access doors (I25).-GHPUs of Cluster 2 may have some restraints due to their lack of procedure for self- and hetero-directed aggressive risk management (I40) as well as the lack of having implemented on a regular basis a staff training program on self- and hetero-aggression risk management (I54).-The No-restraint GHPUs of Cluster 3 and Cluster 4 are both characterized by their outcome assessment practice and an open doors policy, which is in line with the physical characteristics of the GHPU (no armored access door, garden not for the exclusive use of GHPU) and the presence of risk management procedures and training. Cluster 3 distinguishes itself from GHPUs of Cluster 4 following a streamlined open-door-oriented policy in their approach as “In-patients are allowed to go out alone” (I53). They also take care to offer individual therapy in private spaces (I45), and have “no procedure with the local police established (I51).

## 4. Discussion

This research is informed by the principles upheld in the United Nations Convention on the Rights of Persons with Disabilities, and it responds to the Council of Europe’s resolution to an imminent transition for eradicating coercive practice in mental health settings [1,2]. Reflecting on Italy’s progressive reforms in psychiatric treatment, initiated by Franco Basaglia’s movement against institutional and coercive practices, our study examines the implementation of ‘No restraint’ policies within 24 Italian General Hospital Psychiatric Units (GHPUs), or SPDCs. Despite these services being considered a minority of the 323 Italian GHPUs, our analysis demonstrates the feasibility and ethical alignment of operating without restraints. Specifically, our research documented the existence of 14 GHPUs in Italy that reported zero restraint episodes during 2022. Furthermore, 10 GHPUs are committed to reaching the same objective and, in most cases, have obtained encouraging results. It may be assumed that all the GHPUs involved in the study—except for one case that reported 43 restraint episodes—apply mechanical restraints as an exceptional event, not as a current practice. Commitment to the ‘No restraint’ policy is confirmed by the fact that in case a restraint episode occurs, most GHPUs ensure the continuous presence of a professional at the user’s side and recognize that it is a warning event requiring attention.

As illustrated in Figure 1, most GHPUs applying the ‘No restraint’ policies are located in North- and Central Italy, with two notable exceptions in the southern regions. Looking at the geographic distribution, we also observed that except for one case, in all Italian regions, ‘No restraint’ GHPUs co-exist with GHPUs that apply restraint measures. This finding is in line with the lack of homogeneity that has been previously observed among regions and among GHPUs within the same region—a lack of homogeneity that ‘is not justified on epidemiological grounds’ [25]. In fact, significant differences have been reported among regional plans [26] despite the recommendations released by the Conference of Autonomous Regions and Provinces in 2010 [25]. These national recommendations promote, among others, the monitoring of both restraints and violent episodes, as well as appropriate training for people involved. The funding from the Ministry of Health, which was mentioned in the Introduction, is expected to partially fill these gaps [14].

When looking at the localization of the GHPUs, we noticed that all of them are situated in small cities, while no No-restraint GHPU has been documented in metropolitan areas. This observation aligns with previous findings collected in Norway, where higher levels of restraint were registered in wards located in urban areas compared to wards situated in small towns or rural areas [15].

Our study found that GHPUs applying ‘No restraint’ practices are more likely to have implemented structured procedures, particularly in risk management and outcome assessments, highlighting the importance of organizational protocols in reducing coercive methods (see Figure 5). In fact, our data indicate that units reporting zero instances of restraint in 2022 often engage in systematic evaluations. This contrasts with services that report instances of restraint, highlighting a significant opportunity for improvement. These findings are in line with the Six Core Strategies model, which emphasizes the systematic collection and analysis of data on the use of seclusion and restraint to identify patterns, monitor progress, and inform practice changes [27,28,29,30].

Previous research has consistently underscored the therapeutic environment’s paramount role in reducing restraint use within psychiatric settings. Notably, a rigorous multi-baseline study conducted by Borckardt et al. [31] in a state psychiatric hospital, analyzing various components, including leadership, monitoring, staff education, and the therapeutic environment, found that an inviting and calm unit environment plays a crucial role in establishing a conducive atmosphere that benefits both patients and staff. This finding emphasizes that, beyond specific interventions, the overall ambiance and design of the psychiatric care setting can significantly influence patient outcomes and behaviors.

Building on this foundational understanding, our study contributes to the ongoing discussion by identifying a significant positive correlation between the number of restrained users and the presence of a garden or outdoor space for the exclusive use of GHPUs’ users. This observation is in line with a study by van der Schaaf et al. [20], which reported an elevated risk of seclusion associated with the availability of outdoor spaces in locked psychiatric wards; however, it is also in contrast to Oostermeijer et al. [32] which have demonstrated that structural modernizations facilitating access to gardens, recreational areas, and sports facilities correlate with decreased seclusion and restraint. These contrasting findings indicate that various external factors, such as the quality and accessibility of outdoor areas, could similarly affect our results.

Our analysis suggests that garden or outdoor spaces for the exclusive use of GHPUs’ users, although potentially beneficial in creating a more inviting setting, might inadvertently isolate service users from the wider hospital environment. This segregation could intensify feelings of confinement and frustration, possibly leading to an increase in escape attempts and aggressive behaviors. Additionally, this exclusivity could mirror elements of an outdated psychiatric care model, where green spaces, although present, were intentionally structured to prevent entry by anyone other than users and psychiatric professionals, perpetuating a sense of isolation.

Given these insights, future research should investigate the effects of unrestricted access to general hospital gardens and spaces in GHPUs that do not practice restraints and how an effectively supported therapeutic environment influences the dynamics between users and staff. It would also be valuable to compare GHPUs that offer the exclusive outdoor area as the sole option for service users with those that provide various choices, including access to other hospital services, outdoor gardening, and external recreational activities, to further understand the physical environment’s role in enhancing psychiatric care.

Despite these observations, it is possible that the ‘No restraint’ practices may also be influenced by factors not directly measured in this study, such as the overall quality of care, user-staff ratios, or specific therapeutic interventions employed alongside restraint reduction strategies. Additionally, variations in organizational culture, staff attitudes toward coercion, and user populations across different settings could contribute to differences in the implementation and impact of ‘No restraint’ policies. The Safeward Model proposed by Bowers takes into account six different domains that may have an impact on conflict and containment, including the physical environment, the staff team, the outside hospital, the user community, user characteristics, and the regulatory framework [17]. Therefore, staff attitudes and emotions, such as anxiety and frustration or moral commitment, also have a significant influence on the internal structure of the ward [17] and need to be further explored through qualitative research. Furthermore, incorporating users’ and staff’s perspectives can enrich the ethical rigor of such studies. These additional factors highlight the complexity of mental health care environments and suggest that a multifaceted approach is necessary to fully understand and optimize the conditions under which restraint-free care can be most effectively achieved.

### 4.1. Strengths and Limitations

The study’s strength is its provision of a detailed description of the organizational aspects of ‘No restraint’ services within GHPUs in Italy. Through the employment of GLM, Cluster Analysis, and Multiple Correspondence Analysis (MCA), it offers an explorative analysis of the complex interplay among various factors that support and may hinder the implementation of non-restraint practices.

However, the study also has limitations that should be acknowledged. The fact that we monitored physical restraint but did not account for potential pharmacological restraint represents an important confounding variable. We hypothesize that facilities with no-restraint policies adhere to high ethical standards, which likely include minimal use of medications. A study from Ravenna [12] supports this hypothesis, indicating that no-restraint policies are associated with stable medication use. Nonetheless, more studies are needed to confirm these findings and control this confounding variable to accurately assess the impact of no-restraint policies on patient care and resource allocation. Moreover, in reviewing the findings presented by Haines-Delmont et al. [33], we found the examination of medication use during restraint procedures and the methods of administration to be a compelling area for further research.

The cross-sectional design limits the ability to infer causality between policies and organizational characteristics and the ‘No restraint’ practice. Additionally, the reliance on self-reported data from GHPUs may introduce bias or inaccuracies in reporting practices and outcomes. Future research could benefit from longitudinal designs to better ascertain the long-term effects of ‘No restraint’ policies and incorporate direct observations or independent assessments of practices to minimize reporting biases. Furthermore, introducing a ‘Restraint’ control group may help falsify our correlations and/or uncover novel possible correlations.

Finally, the transferability of our findings to other contexts or mental health care systems may be limited by several factors. First, the unique organizational, cultural, and regulatory environment of Italian GHPUs (SPDCs) may not be directly comparable to mental health services in other countries or settings. Additionally, the specific characteristics of the user population, staff training, and the healthcare infrastructure in Italy may influence the implementation and outcomes of ‘No restraint’ policies in ways that might not be replicable elsewhere. Consequently, while our study provides valuable insights, caution should be exercised when generalizing these findings to different contexts or healthcare systems.

### 4.2. Toward a Framework for a No-Restraint Policy in Psychiatric Services

Our study offers a descriptive analysis of the implementation of a no-restraint policy within General Hospital Psychiatric Units (GHPUs), which are integrated into broader community services. This integration underscores the essential role of community-based support in facilitating patient reintegration and ongoing care under no-restraint policies.

The involvement of multidisciplinary teams within these units has been observed to be crucial. Comprising a range of healthcare professionals, these teams are instrumental in delivering comprehensive care, which is key to managing psychiatric care without physical restraints. Moreover, our study points to the importance of continuous professional training in risk management. Adequate training helps healthcare providers manage challenging situations more effectively and is in line with the principles of a no-restraint policy.

Additionally, our findings highlight the usefulness of robust systems for monitoring and evaluating the effectiveness of psychiatric services that adhere to a no-restraint policy. Such systems are necessary to ensure that practices are continually refined based on patient feedback and new insights.

While our observations are specific to GHPUs, the insights gained may be applicable to similar settings. We recommend a patient-centered approach where patients are actively involved in their care plans, promoting respect for their dignity and rights. Furthermore, maintaining high ethical standards and developing clear policies that advocate for non-coercive methods are essential.

## 5. Conclusions

The findings of this study document the existence of no-restraint services and underscore the necessity for ongoing research, the development of policies, and the refinement of no-restraint practices within mental health care. Future research should aim to explore the long-term outcomes of ‘No restraint’ policies through longitudinal studies and expand the scope to include diverse healthcare settings across different cultural and regulatory environments. Our study calls for a more detailed examination of how no-restraint policies affect pharmaceutical use and the broader implications for healthcare resource allocation. Future research could focus on comparative studies that collect direct data on medication use in facilities with or without restraint policies to better clarify the impact of such policies on healthcare resource management and patient well-being. Policymakers should consider these findings to guide the creation of supportive frameworks that encourage the adoption of restraint-free practices, emphasizing the importance of training, organizational culture and mission, and user-centered care. In practice, mental health services should strive to incorporate evidence-based strategies to minimize the use of restraints, thus fostering an environment where user safety and dignity are paramount.

## Figures and Tables

**Figure 1 healthcare-12-01104-f001:**
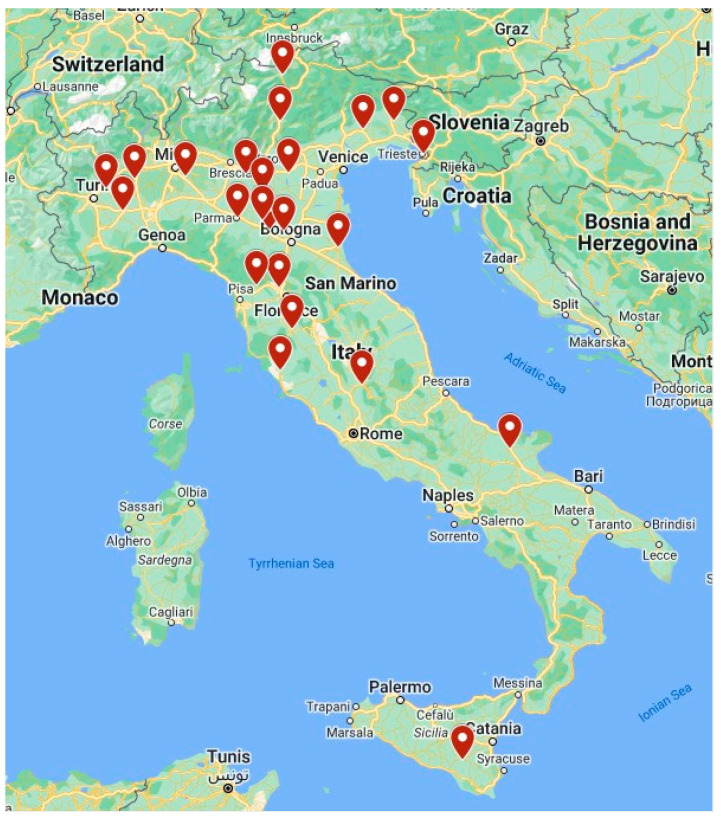
Synopsis of the geographic location of the general hospital psychiatric units (GHPUs) committed to ‘No restraint’.

**Figure 2 healthcare-12-01104-f002:**
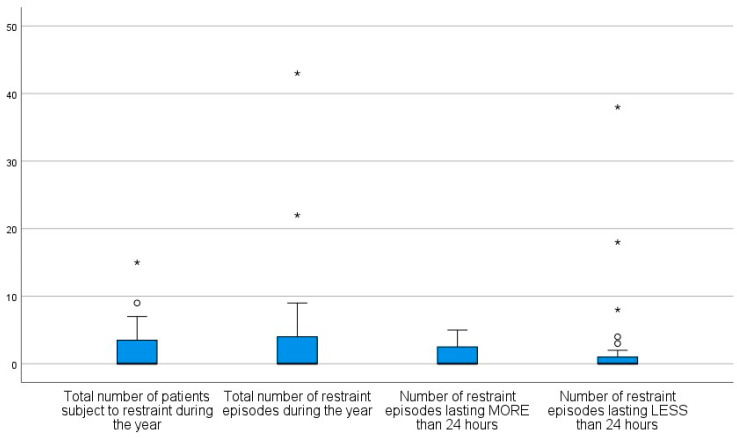
Boxplots of the numbers of—(1) people subject to restraint, (2) restraint episodes, (3) restraint episodes lasting more than 24 h, (4) restraint episodes lasting less than 24 h—in the 24 departments in 2022. (◦ denote mild outliers; * denote extreme outliers).

**Figure 3 healthcare-12-01104-f003:**
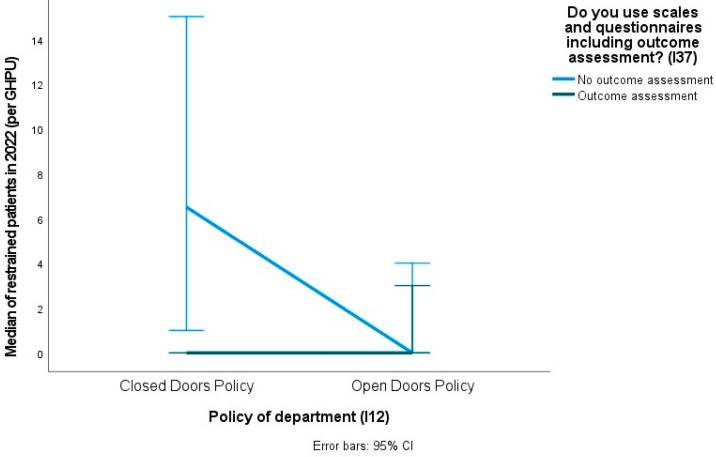
Interaction effect of the presence/absence of an open doors policy and the presence/absence of outcome assessments in the GHPU on the median number of restrained patients.

**Figure 4 healthcare-12-01104-f004:**
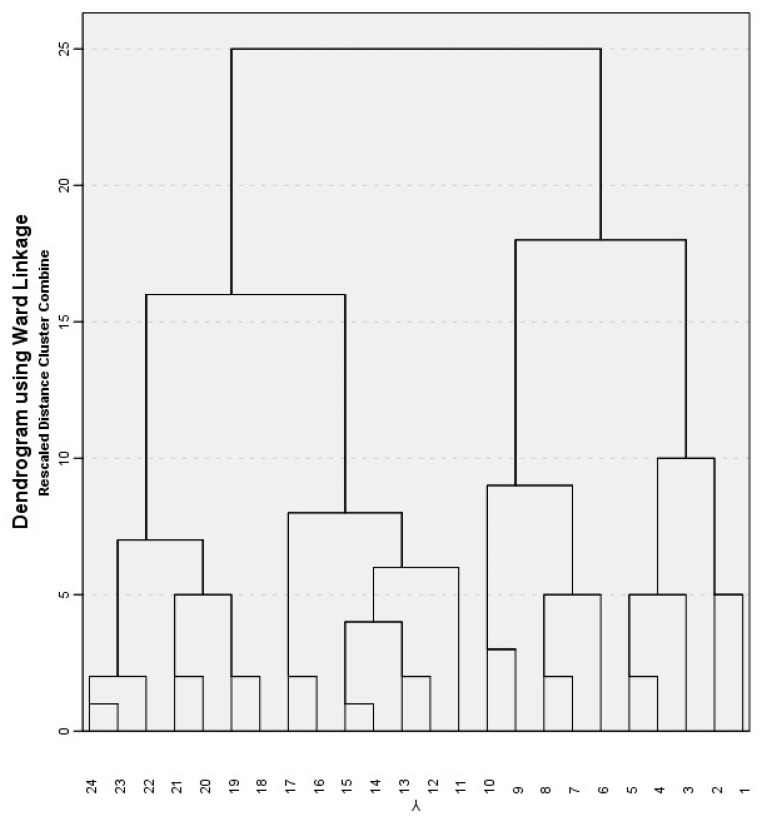
Dendrogram using Ward linkage illustrating the hierarchical clustering process based on which a four- and a two-cluster-solution of the 24 GHPU emerged.

**Figure 5 healthcare-12-01104-f005:**
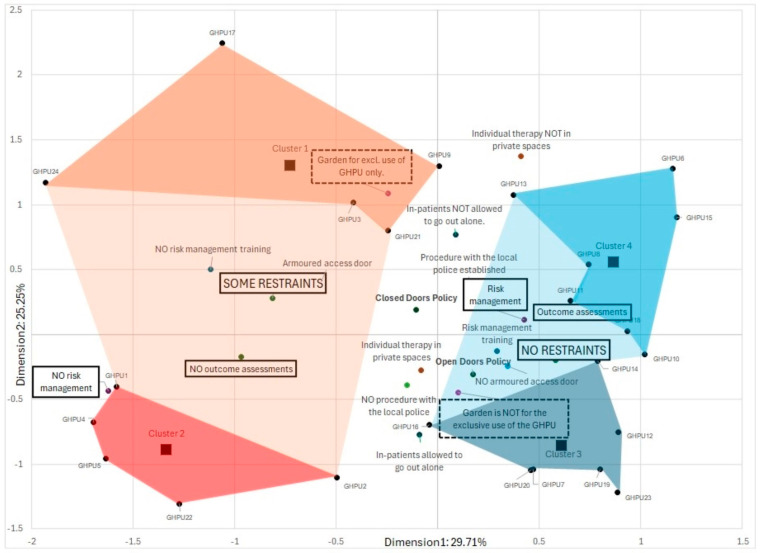
Biplot of Multiple Correspondence Analysis (MCA): Dimension 1: 29.7%; Dimension 2: 25.3%. Note. The biplot of the MCA depicts the clusters of the hierarchical cluster analysis based on the same variables. The reddish- and blue-colored shades cover the area of the GHPUs of a two-cluster solution. Furthermore, the four darker-colored areas distinguish the GHPUs of a four-cluster solution. The labels of categories highlighted with black rectangles indicate the variables significantly associated with no restraint-GHPUs.

**Table 1 healthcare-12-01104-t001:** Garden and outdoor space.

Type of Garden/Outdoor Space	Garden/Outdoor Space Present in Departments
Garden for the exclusive use of GHPU	*n* = 7/24, 29.2%
Other outdoor spaces for the exclusive use of the GHPU	*n* = 5/24, 20.8%
Garden shared with other wards	*n* = 4/24, 16.7%
Other outdoor spaces shared with other wards	*n* = 2/24, 8.3%

**Table 2 healthcare-12-01104-t002:** Professional profile of the staff.

Profession	Employed in GHPUs	Staff Employed Median (Range)
Medical Doctors	*n* = 24/24, 100.0%	4 (1–14)
Nurses	*n* = 24/24, 100.0%	14.5 (9–29)
Psychologists	*n* = 11/24, 45.8%	0 (0–6)
Care Workers	*n* = 22/24, 91.7%	5 (0–12)
Psychiatric Rehabilitation Technicians	*n* = 7/24, 29.2%	0 (0–6)
Social Workers	*n* = 7/24, 29.2%	0 (0–4)
Other	*n* = 8/24, 33.3%	0 (0–2)

**Table 3 healthcare-12-01104-t003:** Shift-wise distribution of professions.

Profession	Morning Shift	Afternoon Shift	Night Shift
Medical Doctors	*n* = 24/24, 100.00%	*n* = 24/24, 100.00%	*n* = 16/24, 66.67%
Nurses	*n* = 24/24, 100.00%	*n* = 24/24, 100.00%	*n* = 23/24, 95.83%
Psychologists	*n* = 7/24, 29.17%	*n* = 3/23, 13.04%	*n* = 0/24, 0.00%
Care Workers	*n* = 22/23, 95.65%	*n* = 20/22, 90.91%	*n* = 14/22, 63.64%
Psychiatric Rehabilitation technicians	*n* = 6/24, 25.00%	*n* = 6/24, 25.00%	*n* = 2/21, 9.52%
Social Workers	*n* = 3/23, 13.04%	*n* = 0/21 = 0.00%	*n* = 0/21, 0.00%
Other	*n* = 9/23, 39.13%	*n* = 5/23, 21.74%	*n* = 0/21, 0.00%

**Table 4 healthcare-12-01104-t004:** Activities carried out within the GHPUs.

Activity	Activity Carried out within GHPUs
Ward rounds	*n* = 13/24, 54.2%
Individual interviews	*n* = 23/24, 95.8%
Individual and group psychological interventions	*n* = 15/24, 62.5%
Psychosocial interventions for family members	*n* = 12/24, 50.0%

**Table 5 healthcare-12-01104-t005:** Procedures to deal with restraints.

Restraint	Present within GHPU
Procedure to eliminate the use of mechanical and spatial restraints present	*n* = 7/10, 70%
Procedure to reduce their risk and duration in case of restraints present	*n* = 5/10, 50%
Restraints are noted in folders in a register	*n* = 10/10, 100%
Restraints are subject to audits or periodic reviews	*n* = 8/10, 80%
Continuous staff assistance is provided in the case of a restraint	*n* = 8/10, 80%

**Table 6 healthcare-12-01104-t006:** Parameter estimates of negative binomial regression using the number of restrained users as a criterion variable.

				95% CI			
Predictor	*B*	*SE B*	*β*	Lower	Upper	*Z*	*p*
(Intercept)	1.44	0.347	4.20	2.19	8.08	4.14	<0.001
Open doors policy	−1.46	0.706	0.23	0.05	0.86	−2.07	0.038
Garden [excl. GHPU]	1.77	0.446	5.88	2.44	15.71	3.98	<0.001
Outcome assessment?	−1.98	0.743	0.14	0.03	0.56	−2.67	0.008
Risk management?	−1.08	0.577	0.34	0.10	1.14	−1.87	0.062
Policy ∗ Outcome assessment?	2.07	1.045	7.94	1.07	65.63	1.98	0.047

Note. Dependent variable: Total number of users subject to restraint during the year 2022. Model: (intercept); Type of Department Policy (Open doors, Closed doors); Presence of a garden for the exclusive use of the GHPU] (Yes, No); Do you use scales and questionnaires, including outcome assessment? (Yes, No); Is there a procedure for self- and hetero-directed aggressive risk management? (Yes, No); N = 24; R^2^ = 0.703; CI = Confidence Interval for β.

**Table 7 healthcare-12-01104-t007:** MCA Model Summary.

Dimension	Cronbach’s Alpha	Total (Eigenvalue)	Inertia	% Variance
1	0.675	2.548	0.255	25.477
2	0.568	2.045	0.205	20.453
Total		4.593	0.459	
Mean	0.627	2.296	0.230	22.965

Note: The mean Cronbach’s Alpha is based on the mean Eigenvalue.

## Data Availability

The datasets analyzed during the current study are not publicly available but are available from the corresponding authors at reasonable request.

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
