# Peer review of "No-Restraint Committed General Hospital Psychiatric Units (SPDCs) in Italy—A Descriptive Organizational Study"

_healthcare, 2024, doi:10.3390/healthcare12111104_

Round 1

Reviewer 1 Report

Comments and Suggestions for Authors

The reviewed article is full of observations and meaningful reflections on key issues in psychiatric care. The authors address an extremely important and rarely addressed topic, making a significant contribution to the global discourse on humanitarian psychiatric care. The analysis of the organisational features and effectiveness of non-coercive practices is fundamental to improving the quality of care for psychiatric patients. However, as the reviewer I would like expresses objections to the method section, particularly the lack of a thorough description of the research sampling process and the adaptation of the research instrument. This is an important issue as an accurate representation of these processes may affect the reliability and understanding of the study results. 

An important point that would have been worth including in the article is a broader discussion of the process of deinstitutionalisation and psychiatric reform in Italy. Such information could be an important theoretical and practical contribution, outlining the directions of change in the field of psychiatric care also beyond national borders. I also see the need for a more detailed discussion of existing services in the absence of constraints in mental health care in Italy and the factors that may affect the quality of these services. This is an important addition that could enrich the discussion and understanding of the issue.

Another point that would be worth highlighting is to propose elements of a future model of psychiatric care compared to existing ones. Such considerations could be a valuable contribution to the development of the field.

Finally, I would like to draw attention to the need to isolate the discussion of theoretical and practical implications, references to existing research and limitations of the study. Such a clear separation of the different elements of the discussion could facilitate the understanding and interpretation of the results.

Despite the above caveats, the paper is an important contribution to the field of psychiatric care and deserves recognition for the effort put in by the authors in compiling it.

Reviewer 2 Report

Comments and Suggestions for Authors

Page 13/18, Line 83-86: How does this effect the data? Specifically, I feel as if the hospitals still actively used restraints in some form, they aren’t a true “No restraint hospital” and this likely effects the ability to draw conclusions about the true effectiveness of “no restraint” practices and policy.

Page 13/18, Line 161: I don't completely agree with this size of a data pool for drawing larger institutional conclusions about healthcare policy or anything else. N is too small and varied in its composition.

Page 13/18, Line 175-176: Re contacting the various survey respondents due to incomplete data sets also creates the potential for bias in the data collection. The actual method of data collection itself lends itself to personal bias and hurts this paper's broad-scale application.

Page 13/18, Line 183: This p-value (unless I'm missing something) is entirely inappropriate.

Page 13/18, Line 344: While I’m certainly not the best statistician, this is a decent R-squared value for this type of study.

Page 13/18, Line 370: , I felt this was an excellent graphic representation of difference in restraint use between closed door and open door policy facilities.

Page 13/18, Line 515: The authors themselves also appear to consciously be aware of the limitations of this study design and the inability to draw strong conclusions from what they have collected. Too many confounding variables.

Page 13/18, Line 538-539: Authors highlighted my previously mentioned concern surrounding bias or inaccuracy in self-reported data.

Page 13/18, Line 545: My personal biggest issue with this paper is the lack of significant data about pharmacological intervention in these "no-restraint" facilities. This really taints the ability to broadly apply any conclusions based on the data in this study, which should have been given in their study design.

Fig 5 is a bit confusing, but the colors might help. 

Comments on the Quality of English Language

Minor Edits needed. 

Round 2

Reviewer 2 Report

Comments and Suggestions for Authors

The reviewers have taken the feedback positively and made the necessary changes.